# The Radiographic Assessment of Furcation Area in Maxillary and Mandibular First Molars while Considering the New Classification of Periodontal Disease

**DOI:** 10.3390/healthcare10081464

**Published:** 2022-08-04

**Authors:** Mohammed Alasqah, Fahad Dhaifallah Alotaibi, Khalid Gufran

**Affiliations:** 1Department of Preventive Dental Sciences, College of Dentistry, Prince Sattam Bin Abdulaziz University, Alkharj 11942, Saudi Arabia; 2Post Graduate Resident in Periodontics, Riyadh Elm University, Riyadh 13244, Saudi Arabia

**Keywords:** furcation involvement, pocket depth, clinical attachment level, gingival recession, smoking, reliability

## Abstract

This study aimed to evaluate the radiographic reliability in the diagnosis of furcation involvement in first molars. A total of 52 subjects were included in the current study. Personal history regarding smoking was recorded and a periodontal examination was performed. Pocket depth (PD), clinical attachment level (CAL), gingival recession, and furcation involvement in all first molars were assessed for each patient. Periodontal staging and grading were evaluated using the new classification of periodontal disease. Class II and Class III furcation classification were more frequently observed in radiographs than the Class I furcation; however, no significant differences were observed. Radiographic observation of the furcation was seen more when PD and CAL were >5 mm in all molars. The presence of gingival recession and its relation to the radiographic assessment did not reveal any statistically significant association (*p* > 0.05) except for tooth #16. The trend of visibility of furcation radiographically was more as the grade of staging was increased. Moreover, the presence of smoking habits and visibility of furcation radiographically did not have any statistical significance. Smoking may not be a factor in the furcation involvement. There is a direct relationship between the staging and grading of the periodontitis and furcation involvement.

## 1. Introduction

Periodontal disease is an immune-inflammatory phenomenon, which is commenced by the dental biofilm and aggravated by the host. It initiates with the involvement of soft tissue, such as gingivitis, and slowly progresses to underlying periodontal support causing loss of connective tissue, alteration in the cementum, and loss of alveolar bone. The loss of inter-radicular bone in the multi-rooted teeth, along with the loss of connective tissue attachment, leads to furcation involvement [1].

Detection of periodontal furcation involvement is a challenge both clinically and radiographically. Nevertheless, the involvement should be detected at an early stage to initiate the required definitive treatment to maintain the prognosis of the tooth favorably and prevent future attachment loss [2]. Clinical diagnosis of furcation requires good skill from the clinicians. A special periodontal probe is available to access the furcation area. The design of the furcation probe is such that it should easily detect the furcation area. However, that does not mean that buccal furcation is easy and can be detected without hassles using only the probe. Thus, there is a clear limitation in the diagnosis of furcation involvement clinically, which broadens the possible options one could diagnose the furcation involvement [3].

Radiographic diagnosis of periodontal disease is the standard way of assessment, which is always performed along with the clinical assessment as an adjunct to confirm the periodontal diagnosis. For screening of periodontal disease, a two-dimensional radiograph is often used, which is considered sufficient if taken in the right way with the proper technique. Periapical radiograph with parallel cone technique or vertical bitewing radiograph is ideal for accurate periodontal bone loss [4]. Though newer radiographic techniques such as magnetic resonance imaging (MRI) and cone-beam computed tomography (CBCT) are available that aid in the diagnostic accuracy of furcation involvement. However, the limitation of using these radiographic techniques for routine clinical practice makes clinicians depend upon the periapical radiograph, bitewing radiograph, and panoramic radiograph [5]. 

Traditional routine radiographs are a two-dimension picture of three-dimension objects. It is considered a standard mode of radiographic assessment in periodontal diagnosis since it is economic, accessible and comprehensible. It has been reported that involvement of furcation is less commonly detected by the clinical examination than the routine radiographs. The prevalence of furcation involvement was 8% in the mandibular molars and 22% in the maxillary molars using the peri-apical radiographs. In contrast, only 9% in the mandibular molar and 3% in the maxillary molars were found to be detected by the clinical examination [6]. Moreover, 40.4%, 43.7% and 54% of furcation involvement were properly identified by the panoramic radiograph, peri-apical radiograph and clinical examination, respectively [7]. Though studies have reported the accuracy of radiographs, the complex anatomical structure of furcation is often considered a limitation with variable results. Thus, the current study aimed to appraise the radiograph reliability in the diagnosis of furcation involvement in all maxillary and mandibular first molars.

## 2. Materials and Methods

Data for this prospective clinical study was collected from the subjects who were seeking periodontal treatment at the postgraduate clinic in Riyadh Elm University during the period from June 2021 to January 2022. The inclusion criteria for this study were: patients aged 18 to 60 years, diagnosed with periodontitis and vital tooth. Whereas, cervical carries, teeth treated with root canal treatment that did not meet the standard clinical principles, broken or mobile tooth, and absence of at least one adjacent tooth were considered exclusion criteria. The sample size was calculated based on the previous study [6] and a total of 52 patients were required to be included in this study. 

All the patients who satisfied the inclusion and exclusion criteria and who were willing to participate and given consent to the study were included in this study. At first, all the patients included were asked to be sited comfortably in the dental chair. After obtaining the relevant dental and medical history, personal history regarding smoking was recorded and a periodontal examination was performed using the prescribed periodontal chart. Different periodontal parameters such as clinical attachment level (CAL), pocket depth (PD), gingival recession and furcation involvement were assessed for each patient. A detailed examination of the full mouth was performed. However, the specific interest of the tooth involved in the study (all the first molars) was examined. All the following periodontal data were documented at buccal and lingual/palatal around each tooth.

PD and CAL were measured using a UNC-15 probe. PD was assessed from the crest of the gingival margin to the base of the pocket and CAL was evaluated from the cementoenamel junction (CEJ) to the base of the pocket. The Association of PD and CAL to the radiographic detection was categorized into ≤5 mm or >5 mm (considering that the furcation trunk length was about 5 mm). Gingival recession was assessed using a periodontal probe and Nabers probe was used to appraise the presence of furcation involvement. Furcation involvement was classified according to the Hamp and coworkers’ classification [8] (Table 1).

Maxillary molars consist of distobuccal, mesiobuccal and palatal roots with mesial, buccal and distal furcation entrances. Moreover, mandibular molars comprise distal and mesial roots with lingual and buccal furcation entrances. Staging and grading of periodontitis were performed following the new classification of periodontal and peri-implant diseases and conditions [9].

Available radiographs were obtained from the patients’ previous records which were taken for the routine periodontal diagnosis; no additional radiographs were taken for this study. The available radiographs were assessed for the presence of furcation and noted in the periodontal assessment form. Hamp’s classification followed both radiographic and clinical classification which classified Class I for absence of bone loss, Class II for grey shade and Class III for complete radiolucency. 

Clinical examination was performed by a single calibrated examiner. Intra-calibration of the examiner was performed. After examining every 10 cases, randomly 2 cases in the list of 10 were examined to find out the accuracy of the furcation grading and to assess the radiographic accuracy of the furcation assessment. 

### Statistical Analysis

The kappa statistics was used for the intra-examiner reliability. The level of kappa agreement categorized as poor, fair, moderate, substantial and near perfect agreement. [10]. Pearson Chi-Square test was used to find the association between the measurements performed radiologically and smoking status and staging and grading periodontitis. Statistical software IBM-SPSS (Version 23, Armonk, NY, USA) was used to analyze the data. The statistical significance was set at *p* < 0.05.

## 3. Results

A total of 52 patients; 33 males and 19 females were included in the current study with the age range of 21–58 (mean 39.6 ± 10.1) years. Kappa statistics showed near-perfect intra-examiner agreement for all the measurements.

### 3.1. Tooth #16

According to the furcation classification, a total of 27, 8, and 3 sites were identified as Class I, Class II, and Class III, respectively. None of the Class I furcation was detected via radiographs and 3 were identified in the radiographs for Class II and Class III furcation. Buccal PD ≤ 5 mm was observed on 23 sites and only two of them were identified radiographically. In addition, 4 out of 15 were identified radiographically when buccal PD > 5 mm. Palatal PD ≤ 5 mm and >5 mm was observed on 22 and 16 sites, respectively. Only, 6 with PD > 5 mm were identified in radiographs. Moreover, there is a significant difference observed with PD identified palatally via radiographs (*p* < 0.05).

When CAL was measured buccally, only 1 for ≤5 mm, and when CAL was >5 mm, 5 were detected radiographically. In palatal CAL (≤5 mm) assessment, none out of 21 was detected radiographically. On the other hand, when the CAL was >5 mm, it was visible radiographically in 5 of 18 sites. Moreover, it was statistically significant (*p* < 0.05).

Gingival recession was found on 10 sites on the buccal side and 7 sites on the palatal side. 5 out of 10 sites on the buccal side and 3 out of 7 sites on the palatal side were detected radiographically. Both buccal and palatal sides showed significant differences (*p* < 0.05).

Staging and grading of periodontitis and radiographic observation revealed that there is an increase in the stage of periodontitis furcation was detected radiographically. However, no significant difference (*p* < 0.05) was observed. Smoking and radiographic detection showed that in the case of smokers it was visible only in 5 out of 25 cases. Similarly in non-smokers, it was visible in 1 out of 27 cases. However, it was statistically not significant (*p* > 0.05) (Table 2).

### 3.2. Tooth #26

According to furcation classification, the number of Class I, Class II, and Class III furcation were observed on 22, 12, and 1 furcation sites, respectively. Among these only 1 site for Class I and Class III, 3 sites for Class II could be detectable radiographically. However, no significant difference was found (*p* > 0.05). When buccal PD was ≤5 mm, 1 of 16 sites was detected radiographically and when the PD was >5 mm, it was visible radiographically in 4 of 18 sites. When palatal PD was ≤5 mm, in 1 of 16 sites, and when the PD was >5 mm in 3 of 17 sites was detected radiographically. However, significant difference failed to determine (*p* < 0.05).

When CAL was ≤5 mm, one site and when CAL was >5 mm, 4 sites were detected radiographically on both buccal and palatal sides. Moreover, it was statistically significant (*p* < 0.05). Bucally and lingually/palatally, 3 out of 9 sites and 2 out of 25 sites of the gingival recession were detected radiographically, respectively, and no statistical difference was detected (*p* > 0.05). 

Staging and grading of periodontitis and radiographic observation revealed that, as there is an increase in the stage of periodontitis furcation was detected radiographically. However, it was not statistically significant (*p* > 0.05). Smoking and radiographic detection showed that in the case of smokers it was visible only in 2 out of 25 cases. Similarly in non-smokers, it was visible in 3 out of 26 cases. However, no distinguished significant difference identified (*p* > 0.05) (Table 3).

### 3.3. Tooth #36

According to furcation classification, the number of Class I, Class II and Class III furcation were observed on 24, 13, and 2 sites, respectively. Among these, 10 Class I and all the Class II and Class III cases were visible radiographically. However, no significant difference was identified (*p* > 0.05). Unlike tooth #16 and #26 it was found that when the PD was >5 mm, radiographic observation of the furcation was seen as more and statistically significant both buccally and lingually (*p* < 0.05). Similarly, it was found that when the CAL was >5 mm, radiographic observation of the furcation was seen more both buccally and lingually. However, statistical significance was found only on the buccal side (*p* < 0.05). 

The presence of gingival recession and its relation to the radiographic assessment did not reveal any statistically significant association (*p* > 0.05). The trend of visibility of furcation radiographically was more as the grade of staging was increased from stage II to stage IV (*p* > 0.05). The presence of smoking habits and visibility of furcation radiographically did not have any statistical significance among the smokers and non-smokers (*p* > 0.05) (Table 4).

### 3.4. Tooth #46

According to furcation classification, the number of Class I, Class II, and Class III furcation was observed on 16, 15, and 5 sites, respectively. Among these, 8 Class I, 13 Class II, and 5 Class III sites were visible radiographically. However, it was statistically not significant (*p* > 0.05). Unlike tooth #16, #26, and #36, it was found that when the PD was >5 mm, radiographic observation of the furcation was seen as more and statistically significant both buccally and lingually or palatally (*p* < 0.05). Similarly, it was found that when the CAL was >5 mm, radiographic observation of the furcation was seen more both buccally and lingually/palatally. However, statistical significance was found only lingually/palatally (*p* < 0.05). 

The presence of gingival recession and its relation to the radiographic assessment did not reveal any statistically significant association (*p* > 0.05). The trend of visibility of furcation radiographically was more as the grade of staging was increased from stage II to stage IV (*p* > 0.05). The presence of smoking habits and visibility of furcation radiographically did not have any statistical significance among the smokers and non-smokers (*p* > 0.05) (Table 5).

## 4. Discussion

The treatment of furcation involvement in the molar tooth considered a challenging task in the field of periodontics [8]. Thus, the accurate diagnosis of the furcation involvement is important as there are different treatment strategies for maxillary and mandibular furcation involvement [11,12]. Current study used Nabers probe for the clinical assessment of furcation which is widely used in diagnosing furcation involvement clinically [13]. Although adjunct radiographic diagnoses via CBCT, MRI, and Radiovisiography (RVG) methods are in place to detect the furcation, routine use of these diagnostic methods is not practical and cost-effective [14]. A previous study compared the different radiographic techniques in cases of furcation involvement and stated that CBCT radiographs should not be used as routine periodontal assessment unless a clear clinical indication [15,16]. Although there are many studies published to date to assess the reliability of various radiographic adjunctive with that of the clinical diagnosis, each study concluded and recommended further studies [17,18,19]. Hence, this current study aimed to assess the reliability of routine periapical radiographs to confirm the clinical diagnosis of furcation is well justified.

The present study involved the detection of furcation only in the first molars. Many previous studies where assessed the reliability of radiographic assessment of furcation involvement taking into consideration all the maxillary and mandibular molars [19,20]. Furcation anatomy, root morphology, root diversion, and other anatomical features may vary from tooth to tooth [21]. Thus, results interpreted as a whole for all the furcation for all molar teeth might not give the true picture of the diagnosis. Hence, the present study would be helpful to interpret the results and considered to be more reliable than when it is involving all the furcation involvement for all molar teeth. The current study used the Hamp and co-workers’ classification for the furcation involvement followed by the previous studies [17,19,20]. Thus, categorizing the furcation based on this classification and co-relating this with the radiographic analysis for reliability seems appropriate. 

In the present study, it was clear that the maxillary furcation (#16 and #26) showed similar observations in terms of radiographic detection. However, in some cases, Class II and Class III were seen radiographically but were not statistically significant. The result for the mandibular furcation (#36 and #46) was marginally different than that of the maxillary furcation with Class I and Class II furcation involvements that could be identified in the radiograph. In the maxillary furcation, Class III sites were comparatively more visible; however, the results were not statistically significant. Graetz et al. found similar results in their study where orthopantomograms and peri-apical radiographs could not identify the Class II furcation. Moreover, Class III furcations were mostly found missing in maxillary arch. The reliability of periapical radiographs considers being poor for many reasons. The presence of palatal root obscures the furcation visibility and shadows the area. It also depends on the root diversion and separation of the root. If the roots are too close, that may obscure the radiolucency in the furcation area and furcation may not be visible. The amount of remaining palatal bone also may hinder the visibility of the furcation [22]. Bragger et al. specified in their study that due to overlapping of anatomical structures and complex anatomy, radiographic assessment of furcation cases was not reliable [23].

The present study showed that diagnosis of furcation involvement by radiographical assessment using the routine radiographs from the patients underestimated while comparing to the clinical assessment of furcation involvements. However, intraoral radiographs play important adjunctive role in the furcation diagnosis [3,24]. Hishikawa et al. suggested similar angulation (−10 to 20 degrees) used for detecting proximal caries is appropriate for the identification of furcation involvement [25].

In the present study, an attempt was made to compare the PD, CAL, and gingival recession to that of the radiographic identification of the furcation. It was seen that when the PD and CAL were >5 mm, there is increased chances of observing the radiographic furcation. However, when the PD and CAL were ≤5 mm, the radiographic visibility of furcation was less. Although there is no direct correlation between the PD, CAL, and gingival recession to the furcation, it is understood that as there is more deepening of the sulcus, there is an increased chance of loss of connective tissue attachment with further progression of periodontal disease and involvement of furcation [26].

As per our knowledge, this is the first study taking into consideration the new classification of periodontal disease. Thus, the present study results become the novice data in this regard. It is seen that as there is an increase in the staging or grading of periodontal disease, there is increased visibility of furcation involvement in the radiographs. As there is a progression of periodontal disease there is an increased chance of furcation involvement and its detection possibility in the radiograph [27]. 

In the present study, one of the parameters used to relate the furcation and its radiographic appearance is the smoking status of the individual. The current study showed furcation involvement in both smokers and non-smokers. Thus, radiographic appearance is not influenced by the smoking status of the individual. Relation to smoking and furcation status in the present study is contradictory to some reported previous studies [28,29]. The difference in the study result could be due to the sample size and classification of smoking status.

No studies are free from limitations and the current study is also not an exception. Assessment of furcation during the surgical assessment is accurate and gives the exact dimension of the furcation. Results of the current study could be improved with the inclusion of surgical exposure along with the clinical and radiographic assessment. The present study used Hamp and co-workers’ classification [8] which was used in the other studies too. However, the need for modification of this furcation classification was felt due to inherent difficulty in distinguishing the furcation, especially Class III. A sub-classification of additional to Class ‘‘II to III’’ according to Walter et al. was added in a previous study for obtaining an accurate clinical diagnosis of furcation [30]. However, Hamp and co-workers’ classification was universally used. Utilizing this classification is more useful to compare the study results to the previous studies. Moreover, using advanced radiographs such as CBCT to compare the results of periapical radiograph for furcation assessment could have helped to further explore the over or underestimation of the furcation through the periapical radiograph.

## 5. Conclusions

Radiographic reliability related to furcation probing is poor with slightly better estimation for the Class III and Class II furcation. The radiographic appearance of the furcation involvement observed more in number when clinical PD and CAL are of >5 mm. Smoking may not be a factor in the furcation involvement clinically or radiographically. There is a direct relationship between the staging and grading of the periodontitis and furcation involvement.

## Figures and Tables

**Table 1 healthcare-10-01464-t001:** Furcation classification.

Classification	Parameter
Class 0	No horizontal loss of periodontal supporting tissue
Class I	≤3 mm of horizontal loss of periodontal supporting tissue
Class II	>3 mm of horizontal loss of periodontal supporting tissue (no through-and-through furcation)
Class III	Through-and-through furcation

**Table 2 healthcare-10-01464-t002:** Relation between the Class of furcation and other periodontal data to the radiographic detectability of furcation for #16.

Periodontal Assessments			Detectible by Radiographs		*p* Value
			Yes (%)	No (%)	
Furcation involvement class [8]		Class I	0 (0)	27 (51.92)	0.074
		Class II	3 (5.77)	5 (9.62)	0.563
		Class III	3 (5.77)	0 (0)	0.132
Pocket Depth	Buccal	≤5	2 (3.85)	21 (40.38)	0.082
		>5	4 (7.69)	11 (21.15)	0.219
	Palatal *	≤5	0 (0)	22 (42.31)	0.024
		>5	6 (11.54)	10 (19.23)	0.044
Clinical Attachement Level	Buccal	≤5	1 (1.92)	19 (36.54)	0.062
		>5	5 (9.62)	13 (25)	0.493
	Palatal *	≤5	0 (0)	21 (40.38)	0.007
		>5	6 (11.54)	11 (21.15)	0.050
Gingival Recession	Buccal *	Yes	5 (9.62)	5 (9.62)	0.042
		No	1 (1.92)	27 (51.92)	0.001
	Palatal *	Yes	3 (5.77)	4 (7.69)	0.051
		No	3 (5.77)	28 (53.85)	0.029
Staging and grading of periodontitis [9]		Stage II Grade B	0 (0)	22 (42.31)	0.056
		Stage III Grade B	1 (1.92)	20 (38.46)	0.059
		Stage III Grade C	3 (5.77)	4 (7.69)	0.932
		Stage IV Grade C	2 (3.85)	0 (0)	0.764
Smoker		Yes	5 (9.62)	20 (38.46)	0.098
		No	1 (1.92)	26 (50)	0.099

*; significant differences (≤0.05), %; percentage.

**Table 3 healthcare-10-01464-t003:** Relation between the Class of furcation and other periodontal data to the radiographic detectability of furcation for #26.

Periodontal Assessments			Detectible by Radiographs		*p* Value
			Yes (%)	No (%)	
Furcation involvement class [8]		Class I	1 (1.92)	21 (40.38)	0.065
		Class II	3 (5.77)	9 (17.31)	0.862
		Class III	1 (1.92)	0 (0)	0.946
Pocket Depth	Buccal	≤5	1 (1.92)	15 (28.85)	0.061
		>5	4 (7.69)	14 (26.92)	0.098
	Palatal	≤5	1 (1.92)	15 (28.85)	0.086
		>5	3 (5.77)	14 (26.92)	0.067
Clinical Attachment Level	Buccal	≤5	1 (1.92)	14 (26.92)	0.076
		>5	4 (7.69)	15 (28.85)	0.124
	Palatal	≤5	1 (1.92)	13 (25)	0.274
		>5	4 (7.69)	16 (30.77)	0.398
Gingival Recession	Buccal	Yes	3 (5.77)	6 (11.54)	0.782
		No	2 (3.85)	23 (44.23)	0.094
	Palatal	Yes	2 (3.85)	6 (11.54)	0.352
		No	3 (5.77)	23 (44.23)	0.093
Staging and grading of periodontitis [9]		Stage II Grade B	1 (1.92)	21 (40.38)	0.058
		Stage III Grade B	2 (3.85)	19 (36.54)	0.067
		Stage III Grade C	1 (1.92)	6 (11.54)	0.074
		Stage IV Grade C	1 (1.92)	1 (1.92)	0.993
Smoker		Yes	2 (3.85)	23 (44.23)	0.190
		No	3 (5.77)	23 (44.23)	0.089

%; percentage.

**Table 4 healthcare-10-01464-t004:** Relation between the Class of furcation and other periodontal data to the radiographic detectability of furcation for #36.

Periodontal Assessments			Detectible by Radiographs		*p* Value
			Yes (%)	No (%)	
Furcation involvement class [8]		Class I	10 (19.23)	14 (26.92)	0.678
		Class II	13 (25)	0 (0)	0.085
		Class III	2 (3.85)	0 (0)	0.773
Pocket depth	Buccal *	≤5	11 (21.15)	12 (23.07)	0.783
		>5	14 (26.92)	2 (3.85)	0.047
	Lingual *	≤5	9 (17.31)	10 (19.23)	0.067
		>5	16 (30.77)	4 (7.69)	0.029
Clinical Attachment Level	Buccal *	≤5	8 (15.38)	11 (21.15)	0.039
		>5	17 (32.69)	3 (5.77)	0.021
	Lingual	≤5	10 (19.23)	10 (19.23)	0.898
		>5	15 (28.85)	4 (7.69)	0.099
Gingival Recession	Buccal	Yes	7 (13.46)	1 (1.92)	0.077
		No	18 (34.62)	13 (25)	0.081
	Lingual	Yes	5 (9.62)	1 (1.92)	0.573
		No	20 (38.46)	13 (25)	0.283
Staging and grading of periodontitis [9]		Stage II Grade B	6 (11.54)	16 (30.77)	0.329
		Stage III Grade B	14 (26.92)	7 (13.46)	0.585
		Stage III Grade C	3 (5.77)	4 (7.69)	0.758
		Stage IV Grade C	2 (3.85)	0 (0)	0.854
Smoker		Yes	11 (21.15)	14 (26.92)	0.953
		No	14 (26.92)	13 (25)	0.785

*; significant differences (≤0.05), %; percentage.

**Table 5 healthcare-10-01464-t005:** Relation between the Class of furcation and other periodontal data to the radiographic detectability of furcation for #46.

Periodontal Assessments			Detectible by Radiographs		*p* Value
			Yes (%)	No (%)	
Furcation involvement class [8]		Class I	8 (15.38)	8 (15.38)	0.870
		Class II	13 (25)	2 (3.85)	0.098
		Class III	5 (9.62)	0 (0)	0.065
Pocket Depth	Buccal *	≤5	9 (17.31)	7 (13.46)	0.049
		>5	19 (36.54)	3 (5.77)	0.037
	Lingual/Palatal *	≤5	6 (11.54)	6 (11.54)	0.093
		>5	22 (42.31)	4 (7.69)	0.022
Clinical Attachment Level	Buccal	≤5	7 (13.46)	5 (9.62)	0.875
		>5	21 (40.38)	5 (9.62)	0.088
	Lingual/Palatal *	≤5	3 (5.77)	6 (11.54)	0.879
		>5	25 (48.08)	4 (7.69)	0.044
Gingival Recession	Buccal	Yes	10 (19.23)	3 (5.77)	0.456
		No	18 (34.62)	7 (13.46)	0.390
	Lingual/Palatal	Yes	8 (15.38)	2 (3.85)	0.580
		No	20 (38.46)	8 (15.38)	0.099
Staging and grading of periodontitis [9]		Stage II Grade B	6 (11.54)	16 (30.77)	0.112
		Stage III Grade B	14 (26.92)	7 (13.46)	0.367
		Stage III Grade C	7 (13.46)	0 (0)	0.099
		Stage IV Grade C	1 (1.92)	1 (1.92)	0.998
Smoker		Yes	13 (25)	12 (23.07)	0.575
		No	15 (28.85)	12 (23.07)	0.397

*; significant differences (≤0.05).

## Data Availability

Not applicable.

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
