# Peer review of "The Radiographic Assessment of Furcation Area in Maxillary and Mandibular First Molars while Considering the New Classification of Periodontal Disease"

_healthcare, 2022, doi:10.3390/healthcare10081464_

Round 1

Reviewer 1 Report

The study is an interesting clinical report that brings applicable results regarding furcation assessment by radiograph examination.

I suggest some minor revision regarding the text. Some words are missing into the sentences, for example:

Line 280:"Radiographical assessment ??? underestimated while comparing to..." 

Line 322: "Furcation is more ??? in clinical PD and CAL..."

Plus, in all tables, the name of the classification adopted should be cited either on the table title or in a table legend.

Author Response

REVIEWER 1    

Line 280:"Radiographical assessment ??? underestimated while comparing to..."

Reply to the reviewer: Thank you so much for your comments. Correction has been done as per your comment.

Line 322: "Furcation is more ??? in clinical PD and CAL..."

Reply to the reviewer: Thank you so much for your comments. Correction has been done as per your comment.

Plus, in all tables, the name of the classification adopted should be cited either on the table title or in a table legend.

Reply to the reviewer: Thank you so much for your comments. Correction has been done as per your comment.

Reviewer 2 Report

Authors provide a study to evaluate the radiographic reliability in the diagnosis of furcation involvement in the first molar. It shows that the radiographic is less sensitive to the Class I furcation. Current study will provide guidance to the clinical practice.

Minor: 

It will be better if the percentage can be added in addition to the counts.

Author Response

REVIEWER 2       

It will be better if the percentage can be added in addition to the counts.

Reply to the reviewer: Thank you so much for your comments. Percentages have been added as per your comment.

Reviewer 3 Report

To author:

Review for Manuscript ID: healthcare-1821893 entitled "The Radiographic Assessment of Furcation Area in Maxillary and Mandibular First Molars while Considering the New Classification of Periodontal Disease”

The manuscript is of interest and has merit for publication. However, there are points that need to be corrected as follows: 

1- keywords: The word “ reliability” has to be added. 

2- line 77: The sample size has to be clarified, how conducted? Based on a previous study or a pilot? The calculation has to be referenced. 

3- Materials and method: Clinical and radiographical figures for the same patient of some cases have to be added. 

3- Result: P value for each chi-square test must be added. 

4- Discussion: The first four paragraphs are more like an introduction and need to be shortened. 

BW, 

Author Response

REVIEWER 3 

  • keywords: The word “ reliability” has to be added.

Reply to the reviewer: Thank you so much for your comments. Keyword has been added as per your comment.

  • line 77: The sample size has to be clarified, how conducted? Based on a previous study or a pilot? The calculation has to be referenced.

Reply to the reviewer: Thank you so much for your comments. Correction has been done as per your comment.

  • Materials and method: Clinical and radiographical figures for the same patient of some cases have to be added.

Reply to the reviewer: Thank you so much for your comments. Unfortunately, the radiological department misplaced the pictorial data and couldn’t retrieve any radio graphical figures for this study.

  • Result: P value for each chi-square test must be added.

Reply to the reviewer: Thank you so much for your comments. Corrections have been done as per your comment.

Discussion: The first four paragraphs are more like an introduction and need to be shortened.

Reply to the reviewer: Thank you so much for your comments. Corrections have been one as per your comment.
